Geographic potential of the world’s largest hornet, Vespa mandarinia Smith (Hymenoptera: Vespidae), worldwide and particularly in North America

http://orcid.org/0000-0001-7442-8593 Nuñez-Penichet Claudia 1 2
http://orcid.org/0000-0003-0701-5398 Osorio-Olvera Luis 2 3 7
http://orcid.org/0000-0002-4146-1634 Gonzalez Victor H. 1 4
http://orcid.org/0000-0002-2611-1767 Cobos Marlon E. 1 2
http://orcid.org/0000-0002-6683-9576 Jiménez Laura 1 2
http://orcid.org/0000-0003-3105-985X DeRaad Devon A. 1 2
Alkishe Abdelghafar 1 2
http://orcid.org/0000-0002-0569-8984 Contreras-Díaz Rusby G. 3 5
http://orcid.org/0000-0002-4371-5415 Nava-Bolaños Angela 2
http://orcid.org/0000-0001-5935-7299 Utsumi Kaera 1
http://orcid.org/0000-0003-4319-9315 Ashraf Uzma 6
http://orcid.org/0000-0001-8513-7804 Adeboje Adeola 1
http://orcid.org/0000-0003-0243-2379 Peterson A. Townsend 1 2
Soberon Jorge 1 2 jsoberon@ku.edu
1 Department of Ecology & Evolutionary Biology, University of Kansas , Lawrence, KS , USA
2 Biodiversity Institute, University of Kansas , Lawrence, KS , USA
3 Departamento de Matemáticas, Facultad de Ciencias, Universidad Nacional Autónoma de México , Ciudad de México, Ciudad de México , Mexico
4 Undergraduate Biology Program, University of Kansas , Lawrence, KS , USA
5 Posgrado en Ciencias Biológicas. Unidad de Posgrado, Universidad Nacional Autónoma de México , Ciudad de México, Ciudad de México , México
6 Department of Environmental Sciences and Policy, Lahore School of Economics , Lahore , Pakistan
7 Departamento de Ecología de la Biodiversidad, Instituto de Ecología, Universidad Nacional Autónoma de México , Ciudad de México , México
Harpur Brock
Electronic publication date: 2021 Jan 13
Publication date: 2021
Volume: 9
Electronic Location ID: e10690
Received 2020 Aug 12; Accepted 2020 Dec 11
Copyright: © 2021 Nuñez-Penichet et al.
Copyright year: 2021
Copyright holder: Nuñez-Penichet et al.
License: This is an open access article distributed under the terms of the Creative Commons Attribution License, which permits unrestricted use, distribution, reproduction and adaptation in any medium and for any purpose provided that it is properly attributed. For attribution, the original author(s), title, publication source (PeerJ) and either DOI or URL of the article must be cited.
License URL: https://creativecommons.org/licenses/by/4.0/

Keywords: Asian giant hornet, Dispersal simulation, Ecological niche modeling, Invasive species, Pollinator threats

Funding: Consejo Nacional de Ciencia y Tecnología 740751, CVU: 368747 Programa de Apoyo a Proyectos de Investigación e Innovación Tecnológica (PAPIIT) Dirección General de Asuntos del Personal Académico (DGAPA) Universidad Nacional Autónoma de México (UNAM) IN116018 Luis Osorio-Olvera was supported by the Consejo Nacional de Ciencia y Tecnología (postdoctoral fellowship number 740751, CVU: 368747) and the Programa de Apoyo a Proyectos de Investigación e Innovación Tecnológica (PAPIIT) - Dirección General de Asuntos del Personal Académico (DGAPA) - Universidad Nacional Autónoma de México (UNAM) (Project IN116018). Rusby G. Contreras-Díaz and Angela Nava-Bolaños were supported by the Programa de Apoyo a Proyectos de Investigación e Innovación Tecnológica (PAPIIT) - Dirección General de Asuntos del Personal Académico (DGAPA) - Universidad Nacional Autónoma de México (UNAM) (Project IN116018). The funders had no role in study design, data collection and analysis, decision to publish, or preparation of the manuscript.

==============================
The Asian giant hornet (AGH, Vespa mandarinia) is the world’s largest hornet, occurring naturally in the Indomalayan region, where it is a voracious predator of pollinating insects including honey bees. In September 2019, a nest of Asian giant hornets was detected outside of Vancouver, British Columbia; multiple individuals were detected in British Columbia and Washington state in 2020; and another nest was found and eradicated in Washington state in November 2020, indicating that the AGH may have successfully wintered in North America. Because hornets tend to spread rapidly and become pests, reliable estimates of the potential invasive range of V. mandarinia in North America are needed to assess likely human and economic impacts, and to guide future eradication attempts. Here, we assess climatic suitability for AGH in North America, and suggest that, without control, this species could establish populations across the Pacific Northwest and much of eastern North America. Predicted suitable areas for AGH in North America overlap broadly with areas where honey production is highest, as well as with species-rich areas for native bumble bees and stingless bees of the genus Melipona in Mexico, highlighting the economic and environmental necessity of controlling this nascent invasion.

Introduction

Invasive species represent major threats to biodiversity, as they can alter ecosystem processes and functions (Pyšek & Richardson, 2010; Vilà et al., 2011), and often contribute to the decline of imperiled species (e.g., Wilcove et al., 1998; Dueñas et al., 2018). The economic damage to agriculture, forestry, and public health, resulting from invasive species totals nearly US $120 billion annually in the United States alone (Pimentel, Zuniga & Morrison, 2005), and more than US $172 million for Canada (Colautti et al., 2006).

Even in the midst of the global uncertainty and socio-economic distress resulting from the COVID-19 pandemic, the recent detection of the Asian Giant Hornet (AGH, Vespa mandarinia Smith, Hymenoptera: Vespidae), in North America (Bérubé, 2020; Wilson et al., 2020), received significant public attention. This social insect is the world’s largest hornet (2.5–4.5 cm body length), and occurs naturally across Asia, including in India, Nepal, Sri Lanka, Vietnam, Taiwan, and Japan, at elevations ranging between 850 and 1,900 m (Matsuura & Sakagami, 1973; Archer, 2008; Smith-Pardo, Carpenter & Kimsey, 2020). As in other temperate-zone social hornet species, annual colonies of the AGH, which may contain up to 500 workers, die at the onset of winter and mated queens overwinter in underground cavities. After emerging in the spring, each queen starts a new colony in a pre-existing cavity, typically in tree roots or an abandoned rodent nest (Archer, 2008). Like other species of Vespa, AGH is a voracious predator of insects, and is notable for preying on bees and other social Hymenoptera. Attacks on honey bee hives occur late in the development of the hornet colony and prior to the emergence of reproductive individuals (males and new queens), the timing of which depends on location (e.g., Matsuura & Sakagami, 1973; Matsuura, 1988; Archer, 2008).

In its native range, AGH attacks several species of bees, some of which have developed sophisticated defense mechanisms against attacks (Ono et al., 1995; Kastberger, Schmelzer & Kranner, 2008; Fujiwara, Sasaki & Washitani, 2016). The best documented, colony-level defense mechanism is in the Asiatic honey bee, Apis cerana Fabricius, which can detect site-marking pheromones released by AGH scouts, and responds by engulfing a single hornet in a ball consisting of up to 500 bees. The heat generated by the vibration of the bees’ flight muscles, and the resulting high levels of CO2 from respiration effectively kill the hornet (Ono et al., 1995; Sugahara, Nishimura & Sakamoto, 2012). In contrast, European honey bees (A. mellifera L.) does not detect and respond to AGH marking pheromones, and colonies are mostly defenseless against AGH attacks (McClenaghan et al., 2019). In Japan, as few as a dozen AGH can destroy a European honey bee colony of up to 30,000 individuals, and extirpate thousands of beehives annually (Matsuura & Sakagami, 1973).

In addition to the potential threat to the beekeeping industry (Alaniz, Carvajal & Vergara, 2020), the introduction of AGH in North America is also concerning for public health. Their powerful stings can induce severe allergic reactions or even death in hypersensitive individuals and sometimes have long-term health effects in people who receive multiple stings (Schmidt et al., 1986; Yanagawa et al., 2007). Annually, as many as 30–40 people may die from AGH stings in Japan, most as a result of anaphylaxis or sudden cardiac arrest (Matsuura & Sakagami, 1973); similar deadly cases have been reported from China during outbreak events (Li et al., 2015). Although invasive species are typically limited by dispersal ability and suitability of novel environments, vespid hornets are well known for their invasive success and excellent dispersal capacity (Beggs et al., 2011; Monceau, Bonnard & Thiéry, 2014). As such, the introduction of AGH in the Pacific Northwest presents a potentially serious ecological and socio-economic risk in North America. Here, we use ecological niche modeling (ENM) to detect areas of suitable environments for this species worldwide, with particular emphasis on North America. We also use a dispersal simulation approach to detect potential invasion paths of this species within North America. A similar methodology for projecting AGH invasion potential has been implemented by Zhu et al. (2020); we build upon this framework by introducing several modifications to the modeling approach, and investigating further the potential ecological impact on the species richness of the genera Bombus and Melipona as well as the possible economic effects of an AGH invasion in North America.

Methods

Occurrence and environmental data

We downloaded occurrence data for V. mandarinia from the Global Biodiversity Information Facility database (GBIF.org, 2020; https://www.gbif.org/). We kept records from the species’ native range (Fig. 1) separate from non-native occurrences facilitated by human introduction. We cleaned occurrences from the native distribution following Cobos et al. (2018) by removing duplicates and records with inconsistent georeferencing (coordinates outside country limits, on the sea, or missing, as recommended in the literature of data cleaning; Chapman, 2005). To avoid model fitting influences of spatial autocorrelation and overdominance of specific regions due to sampling bias, we thinned these records spatially in two ways: by geographic distance and by density of records per country (Fig. 2). In the first case (distance-based thinning; Anderson, 2012), we excluded occurrences that were <50 km away from another occurrence record. In the second thinning approach (country-density thinning), we accounted for potential differences in survey and reporting activities among countries and randomly reduced numbers of occurrences in countries with the densest sampling, namely Japan, Taiwan, and South Korea (from 30, 6, and 5, to 6, 2, and 2 occurrences, respectively), to match the approximate reference density of India, Nepal, and China. We used the package ellipsenm (Cobos et al., 2020; available at https://github.com/marlonecobos/ellipsenm) in R 3.6.2 (R Core Team, 2019) to clean and thin the data. We retained 172 occurrence records for V. mandarinia after initial data cleaning, 49 records after the distance-based thinning approach, and 18 records after the country-density thinning approach (Fig. 1). We then treated both data sets independently in all subsequent analysis steps.

Figure 1 Hypothesis of accessible areas (M) and representation of the occurrence records of Vespa mandarinia across its native distribution.

The three panels represent the occurrences left after cleaning (A) and after applying the two thinning approaches (B and C). Photo credit: Allan Smith-Pardo.

Figure 2 Schematic representation of methods used to obtain ecological niche models for Vespa mandarinia.

The aim of the modeling process was to consider the variability resulting from different procedures and methodological decisions made during model calibration.

For environmental predictors, we used bioclimatic variables at 10′ resolution (~18 km at the Equator) from the MERRAclim database (Vega, Pertierra & Olalla-Tárraga, 2018). We excluded four variables because they are known to contain spatial artifacts as a result of combining temperature and humidity information (Escobar et al., 2014): mean temperature of most humid quarter, mean temperature of least humid quarter, specific humidity mean of warmest quarter, and specific humidity mean of coldest quarter. The 15 variables remaining were masked to an area for model calibration (M, see Ecological niche modeling).

These 15 variables were further processed into two subsets, each used separately in all subsequent analyses. One set consisted of submitting the variables to a principal components analysis (PCA) to reduce dimensionality and multicollinearity. The other set was created with the raw environmental variables that had a Pearson’s correlation coefficient ≤0.85 measured in the calibration area, choosing the most biologically relevant or interpretable variables based on our knowledge of AGH natural history (Simões et al., 2020). As a result of this selection process, we obtained six raw variables: isothermality (BIO3), maximum temperature of warmest month (BIO5), minimum temperature of coldest month (BIO6), temperature annual range (BIO7), specific humidity of most humid month (BIO13), and specific humidity of least humid month (BIO14). The PCA was performed with the raw variables masked to the M area, and principal components for the entire world were obtained by transforming raw variables (at world extent) using the scaling and rotations from the PCA obtained for M. Projecting the PCA results from the accessible area to the entire world ensures that values are comparable and prevents further problems when models are projected. For further analyses, we kept the first four PC axes, as they explained 97.9% of the cumulative variance (Fig. S1). All analyses were done in R; specifically, raster processing was completed using the packages raster (Hijmans et al., 2020), rgeos (Bivand et al., 2020b), and rgdal (Bivand et al., 2020a); PCA was performed using the ntbox package (Osorio-Olvera et al., 2020).

Ecological niche modeling

To identify a calibration area (ostensibly equivalent to M; Owens et al., 2013) for our models, we considered a region contained within a buffer of 500 km around the known occurrence records after the 50 km thinning process (Fig. 1). This buffer distance was selected considering that little is known about the species’ dispersal ability (Matsuura & Sakagami, 1973; APHIS, 2020), as no information exists about queens’ dispersal capacities; however, we consider that given the species’ body size, long-distance dispersal events should not be discarded. We used all pixels in M (15,411) as the background across which to calibrate the models.

Given uncertainty deriving from specific treatments of occurrence records and environmental predictors in ecological niche modeling (Alkishe et al., 2020), we calibrated models via four distinct schemes: (1) using raw variables and distance-based thinned occurrences, (2) using PCs and distance-based thinned occurrences, (3) using raw environmental variables and country-density thinned occurrences, and (4) using PCs and country-density thinned occurrences (Fig. 2). For each scheme, we calibrated models five times, each time randomly selecting 50% of the occurrences for calibrating models (random k-fold evaluation, where k = 5), and using the remaining records for testing (Cobos et al., 2019a).

Each process of model calibration consisted of creating and evaluating candidate models using Maxent (Phillips, Anderson & Schapire, 2006; Phillips et al., 2017). Since many choices are needed to parameterize Maxent (Merow, Smith & Silander, 2013), we chose distinct parameter settings: 10 regularization multiplier values (0.10, 0.25, 0.50, 0.75, 1, 2, 3, 4, 5, 6), eight feature classes (lq, lp, lqp, qp, q, lqpt, lqpth, lqph, where l is linear, q is quadratic, p is product, t is threshold, and h is hinge), and all combinations of more than two predictor variables (Cobos et al., 2019b; Tables S1 and S2). This resulted in a total of 4,560 models using raw variables and 880 using PCs that were tested, in tandem with the two subsets of occurrence data described above. We assessed model performance using partial ROC (for statistical significance; Peterson, Papeş & Soberón, 2008), omission rates (E = 5%, for predictive ability; Anderson, Lew & Peterson, 2003), and Akaike Information Criterion corrected for small sample sizes (AICc, for model complexity; Warren & Seifert, 2011). We selected models with delta AICc ≤2 (Cobos et al., 2019a) from those that were statistically significant and had omission rates below 5%.

After model calibration, we created models with the selected parameter values, using all occurrences after the corresponding thinning process, with 10 bootstrap replicates, cloglog output (Phillips et al., 2017), and model transfers using three types of extrapolation (free extrapolation, extrapolation with clamping, no extrapolation; Owens et al., 2013). Not all calibration processes identified models that met all three criteria of model selection; we did not consider those models in further analyses (Fig. 2; Table 1). As a final evaluation step, we binarized all model replicates given a modified least presence (5% of omission) and then tested whether each replicate of the selected models was able to anticipate the known invasive records of the species in British Columbia, Canada and Washington, USA. For each scheme, using only those model replicates that met the selection criteria and correctly predicted independent occurrences (known invaded localities in North America; independent testing), we created a consensus per sample and two types of final consensus: (1) a median of the medians obtained for each parameterization (continuous), and (2) the sum of all suitable areas derived from binarizing each replicate using a modified least presence (5% omission) threshold (this represents the number of coincidences; Pearson et al., 2007; Fig. 2). Our exploration of different modeling pipelines allowed us to highlight how different methodologies produce different results.

Table 1 Summary of results of ecological niche modeling for Vespa mandarinia, including model calibration, model evaluation, and relevant Maxent settings for models selected after independent testing.

Maxent settings are represented in the columns ‘Regularization multiplier,’ ‘Feature classes,’ and ‘Variable sets’. The column ‘Calibration processes’ refers to each of the five calibrations processes that were done with different sets of random points, for every calibration scheme. The variables included in the sets mentioned on this table can be found in Tables S1 and S2. E: free extrapolation; EC: extrapolation with clamping; NE: no extrapolation, PCs: principal components.

Calibration scheme	Calibration processes	Models meeting selection criteria	Models predicting independent records (E; EC; NE)	Regularization multiplier	Feature classes	Variable sets	
Raw variables and distance thinned occurrences	1	6	8; 2; 10	0.25; 0.5; 0.75	lq; lqpt	42; 43; 50; 51; 57	
2	1	–	–	–	–	
3	1	6; 4; 4	0.75	lqpth	12	
4	1	2; –; 1	0.25	lq	21	
5	2	7; 7; 13	0.1; 0.25	lq	26	
PCs and distance thinned occurrences	1	4	24; 18; 20	5	lqph; lqpth	7; 11	
2	2	10; 5; 5	0.25; 0.5	qp	11	
3	4	22; 23; 19	0.1; 0.25; 0.5; 0.75	lp	11	
4	3	9; 8; 11	0.25; 0.5; 0.75	qp	7	
5	6	21; 21; 26	0.1; 0.25; 0.5; 0.75	lqp	2; 9	
Raw variables and country-density thinned occurrences	1	1	4; 4; 6	0.1	lqp	22	
2	2	4; 11; 8	0.1	lq; lqp	5; 22	
3	–	–	–	–	–	
4	3	15; 13; 16	0.1; 2	lq; lqph; lqpth	13; 32	
5	–	–	–	–	–	
PCs and country-density thinned occurrences	1	7	32; 31; 32	0.25; 0.5; 0.7; 1	lp; lqpt; lqpth	2; 4; 5; 6; 8	
2	1	1; 3; 1	1	lqpth	8	
3	1	5; 6; 7	0.1	q	4	
4	4	24; 22; 18	0.1; 0.25; 0.5; 0.74	lp	1	
5	2	5; 7; 5	1	lqp	1; 6	

As we transferred models to the entire world, we used the mobility-oriented parity metric (MOP; Owens et al., 2013) to detect areas where strict or combinational extrapolation risks could be expected, given the presence of non-analogous conditions with respect to the environments manifested across the calibration area. The areas where extrapolation risks were detected using MOP were deleted from our binary results (suitable areas) to avoid potentially problematic interpretations based on extrapolative situations. Model calibration, production of selected models with replicates, and MOP analyses were done in R using the package kuenm (Cobos et al., 2019a); raster processing and independent testing of models were done using the package raster.

Dispersal simulations

We used the binary outputs from the final consensus models (suitable and unsuitable areas, without areas of strict extrapolation) to simulate invasion dynamics of the AGH. All simulations were started from the Pacific Northwest, from sites already known to be occupied by the AGH. The simulations were performed using the cellular automaton dynamic model included in the bam R package (Osorio-Olvera & Soberón, 2020; available at https://github.com/luismurao/bam). Under this discrete model, given an occupied area at time t, two layers of information are needed to obtain the occupied area at time t + 1: (i) the binary layer of suitability for the species, and (ii) a connectivity matrix determined by the species’ ability to reach neighboring cells in one time unit (known as “Moore’s neighborhood”; Gray et al., 2003, that defines patches that are connected by dispersal). At each step, each of the suitable cells can be either occupied or not by the species. If a cell is occupied, adjacent cells can be visited by the species, and if suitable, they become occupied. This method is similar to the one implemented in the MigClim R package (Engler, Hordijk & Guisan, 2012), but uses a simpler dispersal kernel and parameterization.

With each of the final consensus models for V. mandarinia, we performed a set of simulations in which we explored different degrees of connectivity (1, 2, 4, 8, 10, and 12 neighbor cells pixels per unit time) and different suitability thresholds (10 equidistant levels from 3% to 10% of the presence points to explore variability in sensitivity to the amount of area classified as suitable) to create the binary maps. Since no information is available about dispersal capacities of queens of V. mandarinia, all simulations were done with 200 arbitrary time steps that ensure reaching a steady state. In the end, we visualized the simulation results by summing the occupied distribution layers obtained from each set of simulations. A value of 100 in these final layers means that the species reached that cell in 100% of the simulations, whereas a value of 0 means that the species never reached that cell. Further details regarding the simulation processes can be found in the Supplemental Information.

Honey production and native bee richness in North America

To explore potential ecological and economic impacts of the invasion of the AGH in North America, we explored annual, state-level production of honey (for Mexico, Untied States, and Canada) as well as species richness of bumble bees (Bombus Latreille) and stingless bees of the genus Melipona Illiger in Mexico and the United States. We extracted data on 2016 honey production (in US dollars) for the United States from the U.S. Department of Agriculture (USDA; available at https://quickstats.nass.usda.gov/#4A0314DA-F3E5-3B06-ADD1-CA8032FBD937), from the Instituto Nacional de Estadística, Geografía e Informática (INEGI) for Mexico (https://atlasnacionaldelasabejasmx.github.io/atlas/cap5.html), and from the government of Canada website (https://www.agr.gc.ca/eng/horticulture/horticulture-sector-reports/statistical-overview-of-the-canadian-honey-and-bee-industry-2018/?id=1571143699779) for Canada. For native species richness, we obtained a list of species of bumble bees and stingless bees of the genus Melipona that occur in Mexico and the United States from Discover Life (https://www.discoverlife.org/) and downloaded their occurrence data from GBIF.org (2020). We chose these bee taxa as likely targets of AGH because the species in these groups are of similar body size and behavior to the typical prey of these hornets: they are social insects that form annual or perennial colonies that can have a few hundreds to as many as 10,000 individuals (Cueva del Castillo, Sanabria-Urbán & Serrano-Meneses, 2015; Viana et al., 2015), and store honey and pollen inside their nests (Michener, 2000). To summarize species richness of these two genera, we created a presence-absence matrix (PAM; Arita et al., 2008) for North America, based on geographic coordinates of occurrence data, with a pixel size of one degree. The PAM was created in R with the package biosurvey (Nuñez-Penichet et al., 2020; available at https://github.com/claununez/biosurvey).

To assure transparency and reproducibility of our work, we include an Overview, Data, Model, Assessment, and Prediction protocol (ODMAP; Zurell et al., 2020) in our Supplemental Materials. This metadata summary provides a detailed key to the steps of our analyses. The data and R code used in this research are openly available at http://hdl.handle.net/1808/30602 and https://github.com/townpeterson/vespa repositories, respectively.

Results

Model calibration

The number of models that met the selection criteria was considerably smaller than the total number of models tested (Table 1, see Supplemental Information for more details of the calibration results). The calibration schemes including raw variables had fewer models selected than those using PCs (11, 19, 6, 15 models selected for raw/distance-thinned, PC/distance-thinned, raw/country-density, and PC/country-density, respectively). Not all replicates of selected models anticipated the V. mandarinia invaded areas in North America successfully, so we kept only those that predicted all known invasive records. The number of replicates retained varied among distinct calibration schemes and types of extrapolation used (Table 1).

Ecological niche model predictions

In our models, areas predicted as suitable for the AGH varied among calibration schemes, in both extension and geographic pattern (Fig. 3; Figs. S2–S4). The differences are conspicuous between the two types of thinning approaches, which resulted in models created with different numbers of occurrence records. Models with country-density thinning (18 records) resulted in broad predicted suitable areas worldwide, with areas of higher values of suitability concentrated in tropical regions (Fig. 3; Figs. S2–S4). In contrast, models created with the greater number of occurrences (49 records) from the geographic distance thinning predicted more patches of suitable areas across large extensions of Southeast Asia, Europe, West Africa, Central America, northern South America, and the Pacific Northwest and southeastern United States (Fig. 3; Figs. S2–S4). In the calibration area, the areas detected with high levels of suitability were larger in the scheme with geographic distance thinned occurrences and the raw variables and smaller in the predictions obtained with the country-density thinned occurrences and the PCs as environmental predictors (Fig. 3). In all schemes, the two northernmost occurrence points of this species in China were accorded relatively low levels of suitability (Fig. 3). Predicted suitable areas for this hornet worldwide were also different among types of extrapolation considered in this study, especially as regards its distribution size rather than location (Figs. S2–S4).

Figure 3 Median of potentially suitable areas for Vespa mandarinia predicted with free extrapolation for different calibration schemes in the calibration area (A, C, E and G) and in North America (B, D, F and H).

Only models that anticipated the invaded areas of North America were included. The color pallet is standard for all figure panels.

In North America, across multiple model calibration schemes, our various models agreed in predicting suitable areas for AGH in the Pacific region of southwestern Canada, the Pacific Northwest, the southeastern United States, and from central Mexico south to southernmost Panama (Fig. 4). Our model calibration schemes also agreed in identifying the Rocky Mountains and Great Plains as unsuitable for this species (Fig. 4).

Figure 4 Sum of all suitable areas for Vespa mandarinia in North America derived from binarizing each replicate of selected models (model transfers done with extrapolation), using a 5% threshold and separated by calibration schemes ((A) Raw variables and spatially thinned occurrences; (B) Principal components predictors and spatially thinned occurrences; (C) Raw variables and country-density thinned occurrences; (D) Principal components predictors and country-density thinned occurrences).

Each replicate predicted the known invaded localities of this hornet.

The proportion of area identified as suitable varied among the data thinning schemes. In the case of models created with raw variables, the proportion was 0.171 and 0.164 for spatially thinned and country-density thinned records, respectively. When PCs were used, suitable proportions were 0.248 and 0.239, for spatially thinned and country-density thinned records, respectively (Table S3).

Extrapolation risks in model projections

The pattern of areas detected with risk of extrapolation was similar worldwide between thinning methods, but different between raw variables and PCs (Fig. 5; Fig. S8). Most tropical areas predicted as suitable were identified as regions with high extrapolation risk (Fig. S8). For raw variables, the areas with extrapolation risk in North America included most of Canada and Alaska, whereas for PCs areas with extrapolation risk included large portions of Mexico and, the central-southwestern United States, as well as the islands north of Hudson Bay in Canada (Fig. 5).

Figure 5 Agreement of areas with extrapolation risk for models of Vespa mandarinia in North America, separated by calibration schemes ((A) Raw variables and spatially thinned occurrences; (B) Principal components predictors and spatially thinned occurrences; (C) Raw variables and country-density thinned occurrences; (D) Principal components predictors and country-density thinned occurrences).

Simulations of potential invasion

The simulations of potential sequences of colonization and dispersal of AGH in North America, starting from the known invaded localities, showed agreement among calibration schemes in predicting an invasion across the Pacific Northwest from southernmost Alaska to southernmost California in the United States (Fig. 6). In contrast, we found that the dispersal distance required to invade all the way to the East Coast of North America varied among calibration schemes. In the schemes using raw variables, the route of invasion to reach the East Coast goes from the Pacific Northwest down to California and Mexico, and then up the East Coast of North America. A dispersal distance of 10 cells (where each cell represents ~18 km) was enough to reach the East Coast (see Figs. 6A and 6C). For the scheme using the 50 km spatially-thinned occurrences and PCs, the invasion follows a more direct route from the Pacific Northwest to the East Coast that goes through the United States, and the required dispersal distance to reach the East Coast was only 4 cells (Fig. 6B). Finally, in the case of country-density thinned occurrences and PCs, the invasion goes from the Pacific Northwest through Canada to the Atlantic, and then down the East Coast to the United States. A distance of 8 cells was needed to make this invasion route possible (Fig. 6D).

Figure 6 Results from simulations of the potential dynamics of invasion of Vespa mandarinia in North America.

Dark shades of green show areas that the species reached in a high percentage of scenarios, while light shades of green represent areas reached only rarely by the species. Arrows represent the general path of potential invasion. (A) Raw variables and spatially thinned occurrences; (B) Principal components predictors and spatially thinned occurrences; (C) Raw variables and country-density thinned occurrences; (D) Principal components predictors and country-density thinned occurrences.

Honey production and native bee richness

The areas in North America that our models identified as highly suitable for AGH overlapped broadly with the states where honey production is highest. This overlap was particularly noticeable in southern Mexico and in some states of the Pacific Northwest and eastern US (Fig. 7). We found a similar pattern with the species richness of Bombus and Melipona.

Figure 7 Representation of potential ecological and economic impacts of an invasion of Vespa mandarinia.

(A) Honey production (in US dollars) in Mexico and the United States in 2016. (B) Species richness of the genera Bombus (bumble bees) and Melipona (stingless bees) in North America. The area shaded in gray represents the simulated potential invaded area of Vespa mandarinia in North America obtained with the 50 km spatially thinned occurrences and PCs as environmental predictors. We used this scenario because is the one that best connects the known invaded areas with the eastern United States.

Discussion

The patterns of suitability that we found in North America across multiple input data processing schemes are broadly concurrent with the results obtained by Zhu et al. (2020) and Alaniz, Carvajal & Vergara (2020) (Fig. 6), who used an ensemble modeling approach for the potential invasion of AGH. This concordance with the results of these works (both among our selected models, and between our models and the ensemble models), gives us confidence that the Pacific Northwest and southeastern United States represent suitable areas for AGH. In contrast with the results of Zhu et al. (2020), our dispersal simulations indicate a larger potential invasion area in the United States, with the AGH potentially crossing to eastern North America via a southern invasion route, through Mexico and Texas; a southeast-ward route crossing Idaho, Wyoming, and Colorado; or a northern route across Canada and the Great Lakes region (Fig. 6).

Quantifying the probability of the AGH following any one of the individual dispersal routes presented would require precise quantification of dispersal ability, and discerning the real-world validity of each of the four modeling outcomes. Instead of attempting to guess, we present several models that offer multiple plausible invasion scenarios. Across all scenarios presented, the AGH is expected to establish populations along the coastal Pacific Northwest via short-distance dispersal, and it is likely to invade the southeastern United States if it has even moderate dispersal potential (Fig. 6). It is important to note that these potential invasion routes consider only the natural dispersal ability of this hornet, and do not take into account the effect of potential accidental human-aided dispersal through the transport of soil and wood, where fertilized queen AGHs overwinter (Archer, 1995). Such unwitting human-aided dispersal is a serious concern, as it could potentiate a rapid invasion of this hornet to environmentally suitable, yet currently isolated places across North America. Our simulations allowing AGH to disperse to larger numbers of neighbor cells are perhaps a good illustration of what could be expected if dispersal events to very long distances occur.

Contrasts between our prediction of extensive invasion potential, and Zhu et al. (2020) more conservative predictions, arise from Zhu et al. (2020) use of MigClim (Engler, Hordijk & Guisan, 2012) to model dispersal of the AGH in western North America. MigClim is a cellular automaton platform that models the state of grid cells as occupied or unoccupied. Although we used the same modeling technique, our dispersal kernel is a much simpler “Moore Neighborhood” (Gray et al., 2003) approach, in which cells surrounding an occupied focal cell (to 1, 2…d neighbors) may become occupied, depending on their suitability. MigClim instead assumes a probabilistic contagion model that requires parameter estimates for number of propagules, and short- and long-distance-decay rates. Given the lack of empirical data to inform values for those parameters, we prefer a simpler algorithm to explore how connected clusters of suitable cells are across different values of the single parameter d. Another factor resulting in these differences is the number of simulation steps used in our approach (200). From a biological perspective, this implies that 200 dispersal events resulting in colonization of suitable cells happened. Although this number may appear excessive, it gives a view of a scenario in which no action is taken to prevent AGH invasion in North America and the species builds to large local populations. For a more conservative view of the expected invasion, one could concentrate in areas with high values of suitability on the layers obtained from our simulations.

The areas in North America that our models identified as highly suitable for this hornet overlap broadly with the states where honey production is highest, and species richness of Bombus and Melipona are highest (Fig. 7). These results give credence to public concerns that, if established, the AGH could pose a serious economic threat to the beekeeping industry in Oregon, northern California, Georgia, Alabama, and Florida. In the United States alone, the European honey bee provides at least $15 billion worth of pollination services and generates between $300 and 500 million in harvestable honey and other products each year (Calderone, 2012). Indeed, Alaniz, Carvajal & Vergara (2020) estimate that if spread across the US, the AGH could threaten between 11 and 100 million dollars for hive-derived products and honey bee-pollinated crops production. In Mexico, impacts on the honey bee industry are also expected in tropical areas of the country that have suitable areas for the AGH, particularly in the states of Yucatán, Campeche, and Quintana Roo. Beekeepers in the United States and Mexico may have to adopt mitigation practices to avoid serious losses, such as those developed by Japanese beekeepers including the use of protective screens or traps at the hive entrance that can exclude AGHs based on body size (Matsuura & Sakagami, 1973; Glaiim, Mahdi & Ibrahim, 2008). Potential establishment of the AGH in North America adds an additional layer of environmental and economic stress to a beekeeping industry already suffering from high annual hive mortality rates resulting from the combined effects of pesticides, diseases, and poor nutrition (Goulson et al., 2015).

The ecological impact of AGH on the local bee fauna is more challenging to predict than the economic impact on honey production, because it is not clear which native bee species would be particularly targeted by AGH in North America. We explore Bombus and Melipona species as potential prey candidates of AGH because, among the >4,000 bee species occurring in this region (Ascher & Pickering, 2020), these two groups of bees are social, locally abundant, and make annual or perennial colonies (Michener, 2007; Cueva del Castillo, Sanabria-Urbán & Serrano-Meneses, 2015; Viana et al., 2015). Thus, they may represent predictable food sources for the AGH, particularly in areas where bee colonies remain active year-round. It is crucial to consider this potential threat because both Bombus and Melipona bees are important pollinators that have already experienced population losses and local extirpations, reflecting changes in landscape and agricultural intensification (Brown & Albrecht, 2001; Cameron et al., 2011). Furthermore, these North American bee species are predicted to lack the specialized behavioral responses exhibited by the Asian honey bee, which makes them vulnerable to the attacks of the AGH. The economic and cultural importance of Melipona species in America is well-documented, particularly in the Yucatan Peninsula in Mexico, where these bees have been traditionally raised for honey and were even considered gods outright in Mayan times (Ayala, Gonzalez & Engel, 2013; Quezada-Euán et al., 2018). It is important to mention, however, that the risk to Melipona species may be lower than that to Bombus species because entrances to the hives of some species of Melipona are narrow, allowing a single bee to pass at a time (Couvillon et al., 2007), unlike the entrances to the hives of honey bees and many bumble bees, which are wider.

The AGH is not the first Hymenoptera to invade North America, and species of Vespa are well-known for their invasive success and excellent dispersal capacity (Beggs et al., 2011; Monceau, Bonnard & Thiéry, 2014). The European hornet, Vespa crabro L., a Eurasian species that was accidentally introduced to North America in the 1800s, occupies a range in the United States that encompasses most states east of the Great Plains (Smith-Pardo, Carpenter & Kimsey, 2020). The solitary giant resin bee, Challomegachile sculpturalis is an Asian taxon which was recently introduced in the United States. Only 15 years after its initial detection near Baltimore, Maryland, this species had invaded most of the southeastern United States (Hinojosa-Díaz et al., 2005). These examples indicate considerable precedent for hornet invasion and establishment in the southeastern United States, but the AGH poses a unique biodiversity risk as a direct predator of bees. Because the Pacific Northwest is consistently predicted as suitable for the AGH, preventing further establishment and spread of recently detected introduced populations near Seattle and Vancouver is essential. If these introduced individuals are not eradicated, they may flourish under the suitable climatic conditions, establishing many more colonies that will be difficult to control. Preventing establishment of the AGH in the Pacific Northwest is especially critical because an established AGH population in the Pacific Northwest would provide a source population for potential long-range dispersers that could use multiple potential invasion routes (Fig. 6) to reach suitable habitat in the eastern United States, facilitating full-scale invasion. In light of this, we recommend official monitoring protocols for the vulnerable Pacific Northwest region including on-going citizen science and outreach efforts (https://agr.wa.gov/hornets), which may be the fastest and most effective way to detect potential range expansions.

Although AGH is primarily found in temperate areas in its native range, some of its populations reach subtropical regions like Taiwan (Archer, 2008), which indicates a broad temperature tolerance. This southern part of the species’ native range might explain why our models predicted suitable areas in South America, Africa, and elsewhere (Figs. S2–S7). Although temperature is a critical factor that determines the abundance and distribution of organisms (Sunday, Bates & Dulvy, 2012), factors such as desiccation resistance may be equally important for some species. For example, for ants and some bees, desiccation tolerance is a good predictor of species’ distributions (Bujan, Yanoviak & Kaspari, 2016; Burdine & McCluney, 2019). For example, humidity is important for the regulation of temperature in nests of the European hornet (Klingner et al., 2005) and, in some species of stingless bees, regulation of humidity appears to be more important than regulation of temperature to maintain colony health (Ayton et al., 2016). Unfortunately, heat and desiccation tolerances, factors that might improve predictions of this species’ distributional potential, are unknown for the AGH. In other hornets, subtropical populations tend to have longer population cycles than temperate populations (Archer, 2008), so negative impacts of an AGH invasion may be stronger in tropical or subtropical areas.

In summary, our modeling approach allowed us to recognize how predicted suitable areas can be depending on distinct schemes of data treatment. We showed that this variability can derive from crucial decisions made during the initial steps of ecological niche modeling exercises. These results highlight the importance of such initial decisions, as well as the need to recognize sources of variability in predictions of suitability. This point is of special importance in predicting the potential for expansion of invasive species, as uncertainty increases when models are transferred to areas where environmental conditions are different. Our analyses and simulations revealed the potential of the AGH to invade large areas in North America and the likely paths of such an invasion. We also showed that predicted suitable areas for the AGH overlap broadly with those where honey production is highest in the United States and Mexico, as well as with species-rich areas for bumble bees and stingless bees. These results bring light to the potential implications of uncontrolled dispersal of the AGH to suitable environments in North America, and highlight the need for rapid eradication actions to mitigate potential biodiversity and economic losses.

Supplemental Information

Supplemental Information 1 Sets of environmental predictors obtained with all combinations of two or more raw variables.

BIO3: isothermality, BIO5: maximum temperature of warmest month, BIO6: minimum temperature of coldest month, BIO7: temperature annual range, BIO13: specific humidity of most humid month, BIO14: specific humidity of least humid month.

Click here for additional data file.

Supplemental Information 2 Sets of environmental predictors obtained with all combinations of two or more principal components (PCs).

Click here for additional data file.

Supplemental Information 3 Proportion of suitable areas obtained for Vespa mandarinia in North America, before and after trimming the models with their MOP.

Click here for additional data file.

Supplemental Information 4 Percentage of variance explained for the first 5 principal components.

Click here for additional data file.

Supplemental Information 5 Median of potentially suitable areas for Vespa mandarinia predicted with free extrapolation for different calibration schemes projected to the world.

Click here for additional data file.

Supplemental Information 6 Median of potentially suitable areas for Vespa mandarinia predicted with extrapolation and clamping for different calibration schemes projected to the world.

Click here for additional data file.

Supplemental Information 7 Median of potentially suitable areas for Vespa mandarinia predicted with no extrapolation for different calibration schemes projected to the world.

Click here for additional data file.

Supplemental Information 8 Sum of all suitable areas for Vespa mandarinia derived from binarizing (using a 5% threshold) each replicate of selected models (done with free extrapolation) that predicted the known invaded localities of this hornet.

Click here for additional data file.

Supplemental Information 9 Sum of all suitable areas for Vespa mandarinia in North America derived from binarizing (using a 5% threshold) each replicate of selected models (done with extrapolation and clamping) that predicted the known invaded localities of this hornet.

Click here for additional data file.

Supplemental Information 10 Sum of all suitable areas for Vespa mandarinia in North America derived from binarizing (using a 5% threshold) each replicate of selected models (done with no extrapolation) that predicted the known invaded localities of this hornet.

Click here for additional data file.

Supplemental Information 11 Agreement of areas with extrapolation risk of models of Vespa mandarinia worldwide.

Click here for additional data file.

Supplemental Information 12 Simulations of the likely invasion dynamics of Vespa mandarinia.

Click here for additional data file.

Supplemental Information 13 Overview, Data, Model, Assessment, and Prediction protocol: a detail key steps of our analyses.

Click here for additional data file.

Supplemental Information 14 Overview, Data, Model, Assessment, and Prediction protocol: a detail key steps of our analyses.

Click here for additional data file.

Supplemental Information 15 Result of model calibration processes.

Results of each calibration process is represented. The number at the end of the name of the page represents the calibration replicate number. 50 km: distance thinned occurrences; PC: principal components as environmental predictors; RawVariables: raw variables as environmental predictors; CountryThinning: country-density thinned occurrences; pval_pROC: p-value from partial ROC analysis; AICc: Akaike Information Criteria corrected for small sample sizes; W_AICc: weight of AICc; N_parameters: number of parameters in the model; M: regularization multiplier; F: feature class; l: linear; q: quadratic; p: product; t: threshold; h: hinge. The variables included in the sets mentioned on this table can be found in Tables S1 and S2.

Click here for additional data file.

We would like to thank the members of the KUENM group for their support in the development of this manuscript. We also thank Allan Smith-Pardo for letting us use the photograph of AGH in lateral view (Fig. 1B). ANB would like to thank Secretaría de Educación, Ciencia, Tecnología e Innovación de la Ciudad de México.

Additional Information and Declarations

Competing Interests

Author Contributions

Data Availability

The authors declare that they have no competing interests.

Claudia Nuñez-Penichet conceived and designed the experiments, performed the experiments, analyzed the data, prepared figures and/or tables, authored or reviewed drafts of the paper, and approved the final draft.

Luis Osorio-Olvera conceived and designed the experiments, performed the experiments, analyzed the data, prepared figures and/or tables, authored or reviewed drafts of the paper, and approved the final draft.

Victor H. Gonzalez performed the experiments, analyzed the data, authored or reviewed drafts of the paper, and approved the final draft.

Marlon E. Cobos conceived and designed the experiments, performed the experiments, analyzed the data, prepared figures and/or tables, authored or reviewed drafts of the paper, and approved the final draft.

Laura Jiménez conceived and designed the experiments, performed the experiments, analyzed the data, authored or reviewed drafts of the paper, and approved the final draft.

Devon A. DeRaad performed the experiments, analyzed the data, authored or reviewed drafts of the paper, and approved the final draft.

Abdelghafar Alkishe conceived and designed the experiments, performed the experiments, analyzed the data, authored or reviewed drafts of the paper, and approved the final draft.

Rusby G. Contreras-Díaz performed the experiments, analyzed the data, prepared figures and/or tables, authored or reviewed drafts of the paper, and approved the final draft.

Angela Nava-Bolaños performed the experiments, analyzed the data, authored or reviewed drafts of the paper, and approved the final draft.

Kaera Utsumi performed the experiments, analyzed the data, authored or reviewed drafts of the paper, and approved the final draft.

Uzma Ashraf performed the experiments, analyzed the data, authored or reviewed drafts of the paper, and approved the final draft.

Adeola Adeboje performed the experiments, analyzed the data, authored or reviewed drafts of the paper, and approved the final draft.

A Townsend Peterson conceived and designed the experiments, analyzed the data, authored or reviewed drafts of the paper, and approved the final draft.

Jorge Soberon conceived and designed the experiments, performed the experiments, analyzed the data, authored or reviewed drafts of the paper, and approved the final draft.

The following information was supplied regarding data availability:

Data are available at KU ScholarWorks http://hdl.handle.net/1808/30602.

Code is available at GitHub: https://github.com/townpeterson/vespa.

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
