# Peer review of "Geographic potential of the world’s largest hornet, Vespa mandarinia Smith (Hymenoptera: Vespidae), worldwide and particularly in North America"

_PeerJ, doi:10.7717/peerj.10690_

## Round 0.1 · original submission · Minor Revisions

Thanks for your timely submission.

The reviewers have some minor comments that need to be addressed. Please address these and get a revision back to us.

·

Basic reporting

No comment

Experimental design

The experimental design is very comprehensive: the authors experiment on several aspects of model building and correctly document them. Besides, they are very thorough in the modeling analyses, clearly stating and applying concepts that often times are missed by niche modelers. I have only some small to medium concerns that I express here:

1) Whereas the main objective of the study is clearly defined (i.e., simulating the invasion potential of the AGH hornet in NA), the authors also have other interesting methodological objectives that are not stated as clearly (i.e., assessing the effect of particular modeling decisions/approaches in model variability and uncertainty). You can infer this through the reading, but since it is not explicitly stated, the reader might have difficulties with the flow of the manuscript, having the main objective being constantly "hindered by these methodological details". I totally agree with the authors on the importance of these aspects, showing their effects, and having the study be more than a case study of invasive potential. However, I think they will have much to gain if they state these tangential objectives clearly since the beginning, rephrasing/structuring some of the subheaders so that the reader can clearly follow the main findings.

2) My only medium concern regarding experimental design is with the PCA, which the authors undertake to characterize environmental variability within fewer dimensions. However, this PCA is done within a restricted area and then extrapolated to the whole world. To my understanding, one has to very careful when extrapolating PCA's. If the interest was to reduce dimensionality to the few variables that characterize most of the environmental variation both in the training and testing area of the model, the PCA should have been done considering at least these two areas (if not the whole world, although that entails additional caveats). This is because these areas can differ in the collinearity shown by the variables (which is likely, given the differing longitudes and biomes they represent, and the authors actually show with their MOP analyses). If the PCA is done only within the training environmental data, and then those PCs are interpreted within the testing area, whatever collinearity structure that characterized those PCs might change (or not even be present) in the testing area. Hence the need for summarizing/characterizing environmental variability simultaneously for both training and testing areas.

However, given that the authors also present models using another framework besides the PCs, I don't consider this issue to be a major drawback for the study. Redoing the analyses and assessing whether the general findings hold could be fruitful, but it might be too much to ask at this stage of the manuscript.

What I consider that the authors should do is:
a) justify why they did this analysis as such (there are probably other caveats involved when doing a more expansive PCA like the one I suggest).
b) Clearly state this potential caveat and its potential effects in their findings.

3) One lesser thing that the authors should also mention regarding their design pertains to the particular scheme they used to divide occurrences in training and testing subsets (i.e. random partitioning). Theory and practice agree that, lacking truly independent data for model evaluation, the best next thing is spatially structured cross validation. This is because structuring the data this way reduces the probability of the datasets sharing the same biases, as well as reducing the spatial autocorrelation between training and testing datasets. If training and testing datasets are spatially autocorrelated, it is an easy task for the model to predict the testing dataset, becoming a lenient evaluation (and failing to detect any biases both datasets share). Random partitioning usually results in training and testing datasets sharing the same biases because of their spatial proximity, resulting in models that are either fit to biases or overfit to whatever nuances are present in the datasets. Hence, spatially independent evaluation results in models that are less biased and more general (less overfit to the particularities of the dataset), which is especially desired when models are to be transfered to other places, as was the main objective in this study (see Veloz 2009; Muscarella et al. 2014; Radosavljevic & Anderson 2014; Wenger & Olden 2014; Soley-Guardia et al. 2019).

Luckily, here it seems that the occurrences lay far from one another (which the authors improved by filtering). This likely reduced or eliminated this issue, but I consider that the authors would do well in pointing to how their approach reduced the possibility of obtaining inflated evaluation metrics (for justifying their findings but especially to avoid encouraging the continued usage of a method that has usually been criticized under general circumstances). For instance, many novel users do not even filter their datasets, and then random partitioning results in heavily correlated clusters of training and testing datasets.

Validity of the findings

I salute the authors clear documentation of their procedure and results, which allows for the reproducibility of the study (something often times neglected in modeling studies).

Currently, I think that the validity of the findings lose a bit of momentum because a stronger justification is needed for assessing this invasion potential. Currently, there is no explicit mention of the invasion of this hornet in other parts of the world or its effects (at least not in terms of invasion frequency, establishment rate, or the implications for the environment and the economy). If this information does not exist or such invasions haven´t even occurred or resulted in major impacts, the authors should then tone down their discussion a little bit. In no way are the authors being alarmistic or catastrophic, but they do point out to the importance of monitoring this invasive species in several occasions. To me, this current emphasis lacks justification in the writing, and that is why I suggest that authors "match things up" (either by including additional justification information or being careful in that these statements do not come across as overly stressed).

Additional comments

This is a well-written, interesting, and neatly carried study that uses niche models to address an important ecological question while at the same time studying important modeling aspects. I provide general comments here and detailed ones throughout the manuscript (including tables and figures) that I hope help the authors improve it. I look forward to eventually reading a published version.

Reviewer 2 ·

Basic reporting

no comment

Experimental design

no comment

Validity of the findings

no comment

Additional comments

I have carefully read the manuscript entitled "Geographic potential of the world's largest hornet, Vespa mandarinia Smith (Hymenoptera: Vespidae), worldwide and particularly in North America" and I enjoyed reading the manuscript.

Annotated reviews are not available for download in order to protect the identity of reviewers who chose to remain anonymous.

·

Basic reporting

The manuscript is written well. There is no repetition and I found the descriptions to the point. I have pointed out some areas that need editing in my line-by-line recommendations given below. The references given are generally in line with the authors’ statements. I also found all the figures supplied to be relevant. In reference to the tables, I recommend to provide a table showing the parameters used for the dispersal simulations similar to the way the parameters for the habitat suitability modelling is provided in the first table. The table for parameters used for the dispersal modelling belongs in the article not in the supplementary information section as it is one of the main components of the modelling procedure that contributed immensely to the conclusion the authors reached regarding the potential of AGH in North America.

Experimental design

The fact that the authors used multiple scenarios of various parameter, model and evaluation method combinations is extremely commendable. The use of dispersal simulations to delineate the realized niche of the species additional to what the potential suitability is also one of the very important methodological steps the authors took. Presenting suitability studies with dispersal ability considerations gives better information for those who would like to use such studies for actual applied intervention against invasive species.
Although, I understand that the dispersal parameters not sourced from empirical data. I believe a literature review could provide an idea of how far the AGH can disperse. Therefore, it would be possible to give an idea of what one time step or “t” would represent in the dispersal scheme in the methods section. Is one time step “t” considered a given season? A year? I believe this should be specified with a caveat that the representation is not precise due to the simulation not being based on empirically derived parameter values.
The model is described in sufficient detail so that others can replicate the experiment if needed. If possible, authors should provide the occurrence locations they used to perform their niche modelling.

Validity of the findings

The study is statistically sound. Authors have reported a good level of detail for readers to understand how they undertook the modelling section of the work. The effort taken to address multiple scenarios, which usually entails considering various levels for as many factors as possible is commendable.

Additional comments

I found the study in general and the findings in particular to be timely and important considering that one of the identified families, honey bees, to be targeted by this invasive the Vespa mandarinia is already experiencing great population decline. As I have mentioned a couple of times above, I also found the fact that the authors used multi-scenario modeling framework is very useful as the information about the uncertainty of the results that can be obtained from such scheme is very useful if such studies are to be used for applied invasive species control work on the ground. Information on uncertainty of models will allow end-users to prioritize areas of intervention based on the provided additional information about the potential suitability of different risk areas under protection.
- Include a table that shows parameters used for the dispersals simulation in the main article
- Indicate what time span “t” stands for in the dispersal simulation framework
- A rationale to why country boundary based thinning is proposed in the first place need to be given beyond giving reference to a study that has used the same method. The stark difference in the global suitability prediction between the two thinning methods is beyond what is expected from a variation of niche due to reduction in occurrence numbers (sample size). The ecologically irrelevant use of national boundaries as a reason to thin the occurrence data might have artificially reduced the importance of some sets of ecological conditions because density to an extent indicates importance of the environment.
- Both in case of agreement of suitable areas and agreement of areas with extrapolation risk it seems models using principal component based variables fared better, but I do not see this reflected in the discussion, is there a reason for this?
- S3 Table title: change to ‘predicted prevalence’ instead of ‘prevalence’
- Include occurrence dataset used for modelling, unless there is a data privacy arrangement that the authors have to honor. From their descriptions, it seems occurrence data is sourced from the GBIF, which is freely available to the public. Therefore, I believe it would be great if authors could provide the subset of the occurrence data they used for their modelling, as the actual data that people can download from the GBIF database contain far larger dataset that could have been removed from the set the authors used for their paper. Having the actual set will make replicating the experiment if needed easier.
- Include graphs of tables that show complete model evaluation results as it is difficult to obtain that information from within the body of the text when researchers refer to the publication for further modelling work on the species.

Line by line

L115: “… <50 km away from another locality” this part of the statement is confusing. What dos locality mean in this case, a place name, or another occurrence record? Please describe this more clearly.
L117: While I very much agree with the first thinning procedure the second one is not well justified. Why is an administrative boundary such as a national boundary important to be used as a unit of standardization? Is it based on the notion that the sampling schemes each country uses are different? Was an attempt made to check if sampling intensity was different for the various countries? Otherwise, simply more presence cannot be taken for more sampling effort by default.
L152: Any description why the authors used a 50:50 split for model training.
L251-L253: Is not very clear, please edit for clarity. Is this to mean not all selected models had their replicates selected? Is my understanding that each combination (scenario) has models that fulfill the criteria selected?
L271: “… as regards distribution...” insert ‘it’ between “as” and “regards”
L277: It is difficult to call predicted areas as “prevalence”. “Predicted prevalence” would better describe the proportion of the potential suitable area.
L279-L280: Being rare has a specific ecological meaning, which most certainly is associated with having a narrow ecological niche. When we have a prior knowledge that these records were not rare (relative to their methodologically downsized number -or the records left after thinning), therefore I do not think using “rarefied” is an appropriate way to describe the process. It gives the idea that the thinning process deliberately restricted the environmental range. “Spatially downsized” or “Spatially thinned” would be appropriate. I believe using the latter would be even better because that is what was used at the beginning of the article.

---

## Round 0.2 · Minor Revisions

Thank you for your revised manuscript. It was read by three reviewers. All three of them were satisfied with this revision. There were some minor suggestions requested by the third reviewer. Please address these in full.

I look forward to accepting your manuscript. Very nice work.

·

Basic reporting

Meets criteria

Experimental design

Meets criteria

Validity of the findings

Meets criteria

Additional comments

I read the rebuttal letter and was glad to see that the authors neatly stated replies to all of the comments I made.

Regarding methodological things with which I had had some small to medium issues, the authors do a good job in stating their view and justifying it. Whereas I do not entirely agree with their defense for all instances, it is solid and justifiable, and this is not a black and white debate. The approach they mention has its pro's and con's, as does the one I do. I now see the issue with the PCA's and lack of model discrimination when constructing them with a broad environmental domain. Regarding spatial splits for evaluation, the fact that models need to extrapolate is actually (usually) considered as a tangential benefit of this approach (see Wenger & Oden 2014; Soley-Guardia et al. 2019 Suppl docs). Nevertheless, I also understand the authors critiques, especially the difiiculty of characterizing the niche with a reduced biased dataset (especially for a species with such few recrds like this one).

My suggestion for these methodological approaches was more in the line that the ms could be enriched by mentioning their limitations and justifiyng them more explicitly (as to help more novel users and make them aware that they need not follow your approach as a cookbok). However, I don't consider these explanations as critical (there´s always the conflict with word count and keeping the ideas simple enough for the readers too).

Overall, I salute the authors efforts and consider this version ready for publication.

Reviewer 2 ·

Basic reporting

no comment

Experimental design

no comment

Validity of the findings

no comment

Additional comments

Congratulations to the authors, excellent work.

Annotated reviews are not available for download in order to protect the identity of reviewers who chose to remain anonymous.

Reviewer 4 ·

Basic reporting

This article is moving through a second review. The original article apparently was strongly written, easily interpreted, and presented sound science as submitted. The first round of reviews were also thorough, and concerns identified by the previous reviewers were mostly minor and editorial. The few substantive comments focused on model parameters; of these, they seemed more concerned with justifying the modeling decisions than with the execution and interpretation of those models per se. The rebuttal letter provide by the authors and the changes they made to the manuscript are more than satisfactory, and there is little to be gained by another round of second-guessing the modeling approach. The critical aspect is that the modeling choices are explained clearly, which I find them to be, and that the approach of this research team is thus easily compared with the other two recently published HSMs. Other than failing to acknowledge the paper by Alaniz et al, this manuscript is just fine. Please take a look at my suggestions in the manuscript - and address them :-) - but aside from those this article is ready to be read by the PeerJ audience.

Experimental design

The authors have sufficiently explained their methodological considerations and approach.

Validity of the findings

no comment

Additional comments

Hey, nice work and great job responding to the previous round of reviews! I am sad to say that I've added a few (really, just a very few) comments to the track changes version of your revised manuscript. You will see that the majority of my comments are tiny bits of word-smithing that either clarify some of the cited literature or address a few biological aspects that I think should be treated more gingerly in your narrative. The most critical missing thing is failing to address Alaniz et al.'s paper that came out at about the same time as the Zhu et al. article. The findings are not especially different (although they also present a slightly constrained potential geography compared to some of the predictions in this article), but I think you are in a position to now acknowledge and synthesize three separate analyses that have similar findings. It can actually make this manuscript stand out a little if the authors can work that in.

For the record, the suggestions I have made are to my mind fairly minor. They should be addressed, but I don't think any are significant enough to change the overall impact of this paper, and are not something that will require further review.

Annotated reviews are not available for download in order to protect the identity of reviewers who chose to remain anonymous.

---

## Round 0.3 · accepted · Accept

Thank you for addressing all the minor issues raised by reviewers. I can see that you've made those changes where necessary and I appreciate the effort and time you've put into this work.

With that, I am happy to accept your manuscript. Thank you again.